# Overexpression of p-4EBP1 associates with p-eIF4E and predicts poor prognosis for non-small cell lung cancer patients with resection

**Yaoxiang Tang, Jiadi Luo, Yang Yang, Sile Liu, Hongmei Zheng, Yuting Zhan, Songqing Fan, Qiuyuan Wen** *

Department of Pathology, The Second Xiangya Hospital, Central South University, Changsha, Hunan, China

* wenqiuyuan@csu.edu.cn

**Data Availability Statement:** All relevant data are within the paper, and the minimal data set generated during the current study are available in

## Abstract

Eukaryotic initiation factor 4E (eIF4E) and its phosphorylated form (p-eIF4E) play a crucial role in the protein synthesis, both are under regulation of eIF4E-binding protein 1 (4EBP1) and mitogen-activated protein kinase (MAPK)-interacting kinases (MNKs). This study aims to explore the potential prognostic significance of p-4EBP1 and p-eIF4E in NSCLC patients. The expression of p-4EBP1 and p-eIF4E in NSCLC patients was detected by immunohistochemistry (IHC) staining in tissue microarrays (TMAs) containing 354 NSCLC and 53 non-cancerous lung tissues (Non-CLT). The overexpression percentage of p-4EBP1 and p-eIF4E in lung squamous cell carcinoma (SCC) and adenocarcinoma (ADC) was significantly higher than that of Non-CLT. P-4EBP1 expression in patients with advanced clinical stage was higher than that in early stage. Expression of p-4EBP1 had a positive relationship with p-eIF4E expression both in lung SCC and ADC. NSCLC patients with high expression of p-4EBP1 and p-eIF4E alone or in combination had a lower survival rate than that of other phenotypes. For NSCLC patients, p-4EBP1 is an independent poor prognostic factor as well as clinical stage, LNM and pathological grade. Overexpression of p-4EBP1 and p-eIF4E might be novel prognostic marker for NSCLC, who possesses potential application value for NSCLC targeted therapy.

## Introduction

Globally, lung cancer is still the chief reason of cancer morbidity and mortality, its incidence and caused deaths continue to rise, accounting for nearly 20% of cancer deaths [1]. About 85% of lung cancer patients were diagnosed as non-small cell lung cancer (NSCLC) which is a main type of lung cancer [2]. NSCLC can be further divided into two predominant histological subtypes: lung adenocarcinoma (ADC) and lung squamous cell carcinoma (SCC), accounting for more than 70% [2,3]. Clinically, more than 60% of NSCLC patients have advanced disease or metastatic lesions at the primary diagnosis [4]. Although significant progress has been made in NSCLC treatment, its five-year survival rate is still not satisfactory [5]. Therefore, it is urgent to identify new molecular biomarkers to predict the prognosis of NSCLC patients and develop novel molecular targeted therapy.

the figshare repository, https://doi.org/10.6084/m9.figshare.19344524.v1.

**Funding:** The work was supported by National Natural Science Foundation of China (grant No. 81703009) and the Natural Science Foundation of Hunan Province (grant No. 2017JJ3457). The funder is Qiuyuan Wen, who conceived and designed the study and reviewed and edited the manuscript. The work was also supported by National Natural Science Foundation of China (grant No. 81773218 and 81972838). The funder is Songqing Fan, who conceived and designed the study. The work was also supported, in part, by Hunan Provincial Innovation Foundation for Postgraduate (grant No. CX20210373), and the Fundamental Research Funds for the Central Universities of Central South University (grant No. 2021zzts1045). The funder is Yaoxiang Tang, who performed the experiments and wrote the paper.

**Competing interests:** The authors have declared that no competing interests exist.

Cancers are characterized by rapid growth and metabolism and require a high level of protein synthesis (or mRNA translation) [6]. Eukaryotic initiation factor 4E (eIF4E) mediated recognition of 5' cap structure of mRNA is considered as a rate-limiting factor in translation initiation [7,8] by regulating the synthesis of several carcinogenic proteins such as survivin, c-myc, and cyclin D1 [9]. In a mouse model, although reduction of eIF4E expression does not affect the development, it significantly impedes oncogenic transformation [10]. EIF4E-binding protein 1 (4EBP1), as a substrate of mechanistic target of rapamycin (mTOR) signaling pathway, is the main regulator of eIF4E availability [11]. In hypo-phosphorylated state, 4EBP1 competes with eIF4G for binding to eIF4E, thus resulting in the inhibition of translation initiation complex formation [12]. When it is phosphorylated by upstream signals, mainly mTOR complex 1 (mTORC1), eIF4E will be released and drive the synthesis of diverse proteins, including carcinogenic proteins [9,13]. Some researchers have pointed out that high phosphorylated 4EBP1 (p-4EBP1) expression was related with poor prognosis in many major types of cancers, such as renal cell carcinoma, ovarian cancer, and small cell lung cancer [14–16]. In addition, eIF4E can be phosphorylated through the mitogen-activated protein kinase (MAPK)-interacting kinases (MNKs) pathway [17]. Phosphorylated eIF4E (p-eIF4E) has been proved to be overexpressed in many cancers, and be of vital importance in the cancer cell migration/invasion and tumor progression, though its specific mechanism is still elucidated [17,18].

To date, although increasing evidences have demonstrated that p-4EBP1 and p-eIF4E play a critical role in cancer, there are limited data evaluating their importance to NSCLC. To get more insight on the biological significance of these two proteins, we have investigated their expression pattern in tissue microarray (TMAs) including NSCLC specimens and non-cancerous lung tissues (Non-CLT) via immunohistochemistry (IHC), and discussed the internal relationship between p-4EBP1 and p-eIF4E and clinical/pathological/prognostic features in NSCLC.

## Materials and methods

### Patient information and tumor samples

Investigation of p-4EBP1 and p-eIF4E expression in NSCLC was carried out on TMAs including 354 NSCLC cases and 53 cases of Non-CLT received surgical treatment in the Second Xiangya Hospital in Changsha between 2003 and 2013. No patients accepted chemotherapy or radiotherapy prior to surgery. All tumor specimens and 53 samples of Non-CLT acquire from the Pathology Department, the Second Xiangya Hospital of Central South University. Patients in the study had definite histological diagnosis on the basis of 2015 WHO Classification of lung cancer, and were comprehensively staged as per the Eighth Edition Lung Cancer Stage Classification [19]. Clinical information obtained as part of the study included patient age, gender, lymph node metastasis (LNM), clinical stage, histological type, and pathological grade (Table 1). In addition, samples obtained informed consent and approval of all protocols were obtained from the Institutional Human Experiment and Ethics Committee of the Second Xiangya Hospital of Central South University (approval No. S039/2011).

### Construction of the tissue microarrays

High-throughput NSCLC TMAs were constructed as described previously [20]. The perforation diameter of each sample was 0.6 mm. Each NSCLC case included three tumor cores in TMAs and three normal lung cores per case was included in TMAs of Non-CLT.

**Table 1. Clinicopathological features of patients with NSCLC and non-cancerous lung tissues.**

| Patients' characteristics | No. of patients (%) |
|---|---|
| **NSCLC** | |
| **Age(years)** | |
| ≤50 | 98(27.7) |
| >50 | 256(72.3) |
| **Gender** | |
| Male | 268(75.7) |
| Female | 86(24.3) |
| **Clinical stages** | |
| Stage $_\mathrm{I}$ | 78(22.0) |
| Stage $_\mathrm{II}$ | 72(20.4) |
| Stage $_\mathrm{III}$ | 204(57.6) |
| **Lymph node status** | |
| N0 | 142(40.1) |
| N1/N2/N3 | 212(59.9) |
| **Histological type** | |
| SCC | 159(44.9) |
| ADC | 195(55.1) |
| **Pathological grade** | |
| Well | 6(1.7) |
| Moderate | 144(40.7) |
| Poor | 204(57.6) |
| **Non-cancerous lung tissues** | |
| **Age(years)** | |
| ≤50 | 22(41.5) |
| >50 | 31(58.5) |
| **Gender** | |
| Male | 27(50.9) |
| Female | 26(49.1) |

## Immunohistochemistry

Slides from TMAs were stained with anti-p-4EBP1 and anti-p-eIF4E antibodies. The condition of each antibody staining was adjusted as has been shown before [21]. Primary antibodies applied in this study are: rabbit anti-p-4EBP1 (Catalog: #2855, Cell Signaling Technology; 1:800 dilution) and rabbit anti-p-eIF4E (Catalog: #9274, Cell Signaling Technology; 1:500 dilution). The positive control slide and negative control slide were included in each experiment. The IgG isotype-matched antibody was applied as negative contrast to confirm the antibody specificity.

IHC staining was scored independently under x200 magnification light microscopy by two pathologists (Qiuyuan Wen and Songqing Fan). The percentage of positive stained cells were scored as 0~4 according to the following standard: 0 (0%), 1 (1–25%), 2 (26–50%), 3 (51–75%) and 4 (76–100%). The staining intensity of p-4EBP1 and p-eIF4E was divided into four grades: negative, mild, moderate and strong, with scores of 0, 1, 2 and 3 respectively. The total score was based upon semi-quantitative approach, as described below: total score = percentage score + intensity score. According to the overall survival (OS) of NSCLC patients, the best cutoff levels for p-4EBP1 and p-eIF4E expression are 3 and 4 respectively. P-4EBP1 and p-eIF4E were considered to be highly expressed when their total scores were over 3 and 4 respectively,

whereas conversely, they were considered to be low expression. Agreement between the two assessors is 95%, and all the differences in scores are solved through discussion.

## Statistical analysis

SPSS 24.0 (IBM, USA) for Windows was used for statistical analyses. Specifically speaking, chi-square ($\chi^2$) test was used for expression pattern of p-4EBP1 and p-eIF4E between Non-CLT and NSCLC and their association with clinicopathological features (except pathological grade). The association between these two proteins and pathological grade were analyzed via Fisher-exact test. The relationship between p-4EBP1 and p-eIF4E was assessed via spearman's rank correlation coefficient. In addition, survival rate curves were appraised by Kaplan-Meier analysis, and comparisons were analyzed via log-rank test. Furthermore, cox proportional-hazards model was performed for determining independent prognostic markers. $P < 0.05$ (Two-sided) indicates that the result is statistically significant.

## Results

### P-4EBP1 and p-eIF4E were over expressed remarkably in NSCLC

In order to find out whether the level of p-4EBP1 and p-eIF4E proteins in NSCLC is different from that in Non-CLT, we detected the protein expression pattern and cellular localization of them in Non-CLT and NSCLC tissues. Both proteins mainly existed in cell cytoplasm and were positive in lung SCC and ADC (Fig 1A, 1B, 1D and 1E), but negative in Non-CLT (Fig 1C and 1F). According to the histologic subtypes, the percentage of high expressed p-4EBP1 and p-eIF4E proteins was 50.3% (80/159) and 42.1% (67/159) in lung SCC, 43.6% (85/195) and 61.5% (120/195) in lung ADC, respectively. For Non-CLT, both p-4EBP1 protein and p-eIF4E protein showed the same 15.1% positive rate (8/53). As shown in Fig 2, p-4EBP1 proteins in lung SCC and ADC tissues were significantly upregulated, as well as p-eIF4E protein (both $P < 0.001$).

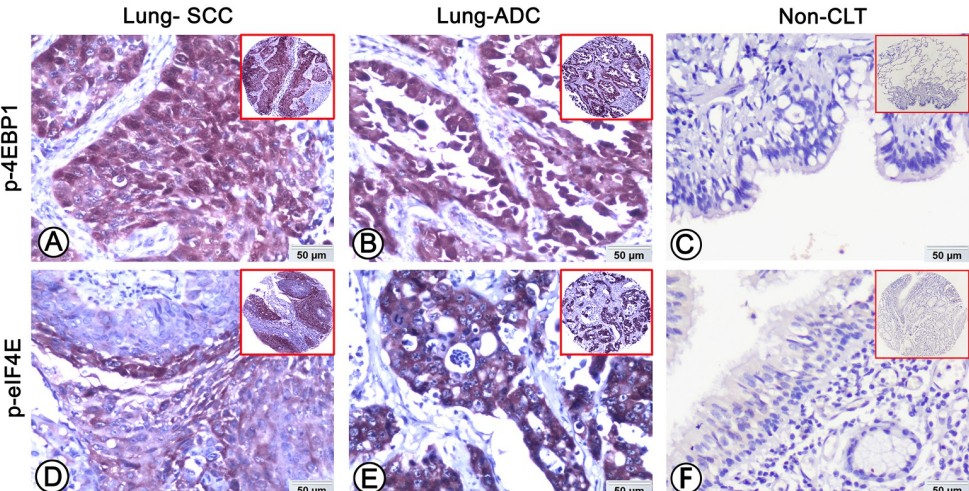

**Fig 1. Expression of p-4EBP1 and p-eIF4E in lung ADC, lung SCC and Non-CLT were detected by IHC.** Strong positive staining of p-4EBP1 (A), p-eIF4E (D) was found in cell cytoplasm of lung SCC cells. Strong positive staining of p-4EBP1 (B), p-eIF4E (E) was also showed in cell cytoplasm of lung ADC cells. Negative staining of p-4EBP1 (C), p-eIF4E (F) was found in Non-CLT (200×, IHC, DAB staining).

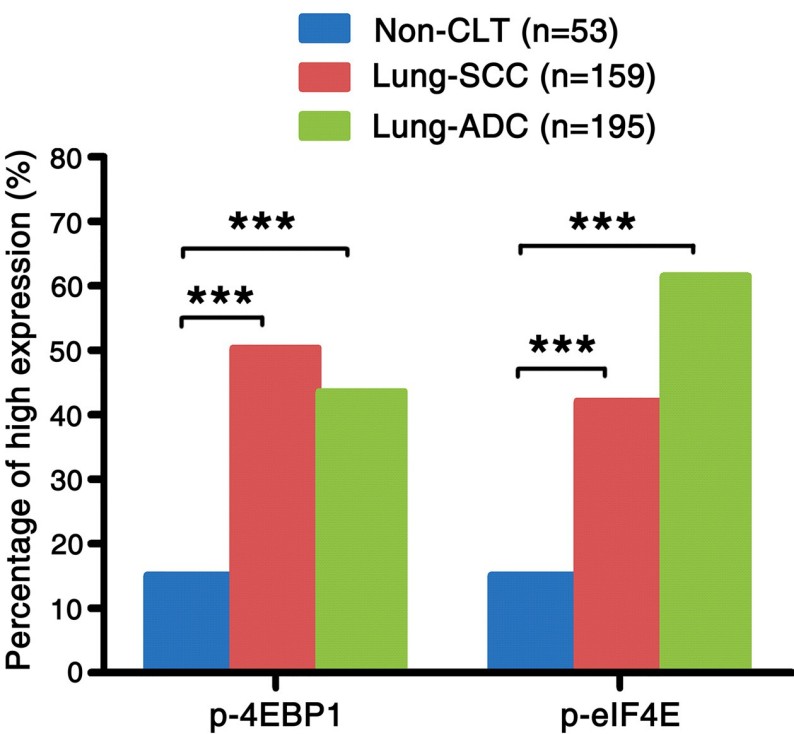

**Fig 2. The comparison of expression of p-4EBP1, p-eIF4E in lung SCC and lung ADC compared to the Non-CLT.**
The expression of p-4EBP1, p-eIF4E in lung SCC and lung ADC was significantly higher than those in Non-CLT (all
$P < 0.001$).

### Relationship between p-4EBP1 and p-eIF4E and the clinical/pathological features of NSCLC patients

We make further efforts to investigate the relationship between overexpressed p-4EBP1 and p-eIF4E and clinical and pathological features. Table 2 showed a positively correlated relationship between overexpression of p-4EBP1 and clinical stages in NSCLC. Compared with clinical stage I and II, the expression of p-4EBP1 in stage III was higher (P = 0.040). Lung ADC patients had significantly overexpressed p-eIF4E than lung SCC patients (P = 0.002). Furthermore, OS rate in patients with individual or combined high expressed p-4EBP1 and p-eIF4E proteins was lower than patients with other phenotypes of them (all $P < 0.01$). No difference was found between p-4EBP1 and p-eIF4E and other clinical/pathological characteristics, such as gender, age, pathological grades and LNM status of patients (all $P > 0.05$).

### Correlation analysis of p-4EBP1 and p-eIF4E proteins expression in NSCLC

Table 3 showed the correlation between increased p-4EBP1 and p-eIF4E proteins in NSCLC. It suggested that overexpression of p-4EBP1 was strong positively associated with p-eIF4E protein both in lung SCC and ADC (r = 0.288, $P < 0.001$; r = 0.397, $P < 0.001$, respectively).

### p-4EBP1 and p-eIF4E expression affect prognosis of NSCLC patients

The survival curve of NSCLC patients was described via Kaplan-Meier method and comparisons of OS rate were performed via log-rank test in the univariate survival analysis. According

**Table 2. Analysis of the association between expression of p-4EBP1 and p-eIF4E and clinicopathological features of NSCLC (n = 354).**

| Clinicopathological features (n) | p-4EBP1 | | | p-eIF4E | | | p-4EBP1 / p-eIF4E [#] | | |
|---|---|---|---|---|---|---|---|---|---|
| | High (%) | Low (%) | P-value | High (%) | Low (%) | P-value | P+ (%) | N- (%) | P-value |
| **Age(years)** | | | | | | | | | |
| ≤50 (n = 98) | 46(46.9) | 52(53.1) | | 48(49.0) | 50(51.0) | | 34(34.7) | 64(65.3) | |
| >50 (n = 256) | 119(46.5) | 137(53.5) | 1.000 | 139(54.3) | 117(45.7) | 0.406 | 82(32.0) | 174(68.0) | 0.704 |
| **Gender** | | | | | | | | | |
| Male(n = 268) | 132(49.3) | 136(50.7) | | 139(51.9) | 129(48.1) | | 87(32.5) | 181(67.5) | |
| Female(n = 86) | 33(38.4) | 53(61.6) | 0.083 | 48(55.8) | 38(44.2) | 0.537 | 29(33.7) | 57(66.3) | 0.895 |
| **Clinical stages** | | | | | | | | | |
| Stage I-II (n = 150) | 60(40.0) | 90(60.0) | | 79(52.7) | 71(47.3) | | 42(28.0) | 108(72.0) | |
| Stage III (n = 204) | 105(51.5) | 99(48.5) | 0.040* | 108(52.9) | 96(47.1) | 1.000 | 74(36.3) | 130(63.7) | 0.110 |
| **LN status** | | | | | | | | | |
| LNM (n = 212) | 102(48.1) | 110(51.9) | | 116(54.7) | 96(45.3) | | 74(34.9) | 138(65.1) | |
| No LNM (n = 142) | 63(44.4) | 79(55.6) | 0.515 | 71(50.0) | 71(50.0) | 0.388 | 42(29.6) | 100(70.4) | 0.302 |
| **Histological type** | | | | | | | | | |
| SCC(N = 159) | 80(50.3) | 79(49.7) | | 67(42.1) | 92(57.9) | | 45(28.3) | 114(71.7) | |
| ADC(N = 195) | 85(43.6) | 110(56.4) | 0.239 | 120(61.5) | 75(38.5) | 0.000* | 71(36.4) | 124(63.6) | 0.112 |
| **Pathological grade** | | | | | | | | | |
| Well(n = 6) | 3(50.0) | 3(50.0) | | 5(83.3) | 1(16.7) | | 3(50.0) | 3(50.0) | |
| Moderate(n = 144) | 60(41.7) | 84(58.3) | | 72(50.0) | 72(50.0) | | 43(29.9) | 101(70.1) | |
| Poor (n = 204) | 102(50.0) | 102(50.0) | 0.292 | 110(53.9) | 94(46.1) | 0.268 | 70(34.3) | 134(65.7) | 0.407 |
| **Survival status** | | | | | | | | | |
| Alive(n = 187) | 68(36.4) | 119(63.6) | | 85(45.5) | 102(54.5) | | 44(23.5) | 143(76.5) | |
| Dead(n = 167) | 97(58.1) | 70(41.9) | 0.000* | 102(61.1) | 65(38.9) | 0.004* | 72(43.1) | 95(56.9) | 0.000* |

*Chi-square test (pathological grade was analyzed via Fisher-exact test), statistically significant difference (P < 0.05).

Abbreviations: LNM, lymph node metastasis; SCC, squamous cell carcinoma; p-4EBP1/p-eIF4E[#] P+, common high expression of p-4EBP1 and p-eIF4E, P-, other combination of expression of these two proteins.

**Table 3. The pairwise association between expression of p-4EBP1, and p-eIF4E in lung SCC and ADC.**

| | SCC | | ADC | |
|---|---|---|---|---|
| | p-4EBP1 | p-eIF4E | p-4EBP1 | p-eIF4E |
| p-4EBP1 | | | | |
| r | 1 | 0.288 | 1 | 0.397 |
| Sig. (2-tailed) | | 0.000* | | 0.000* |

*Spearman's rank correlation test, statistically significant difference. * P < 0.05. SCC, squamous cell carcinoma; ADC, adenocarcinoma.

to the Fig 3, the survival rate of NSCLC patients with low expressed p-4EBP1 (*P* < 0.001, Fig 3A & Table 4) and p-eIF4E (*P* = 0.037, Fig 3B & Table 4) was higher than patients with high level of above proteins respectively. NSCLC patients with high expressed p-4EBP1 and p-eIF4E proteins had shorter survival time than that of other patterns (*P* = 0.003, Fig 3C). We gone a step further and analyzed the relationship between OS rate and clinical and pathological characteristics. The survival rate of NSCLC patients with clinical stage III was lower compared to stage I and II (*P* < 0.001, Fig 3D & Table 4), and patients without LNM had higher survival rate (*P* < 0.001, Fig 3E & Table 4). As for NSCLC patients, the higher the pathological grade

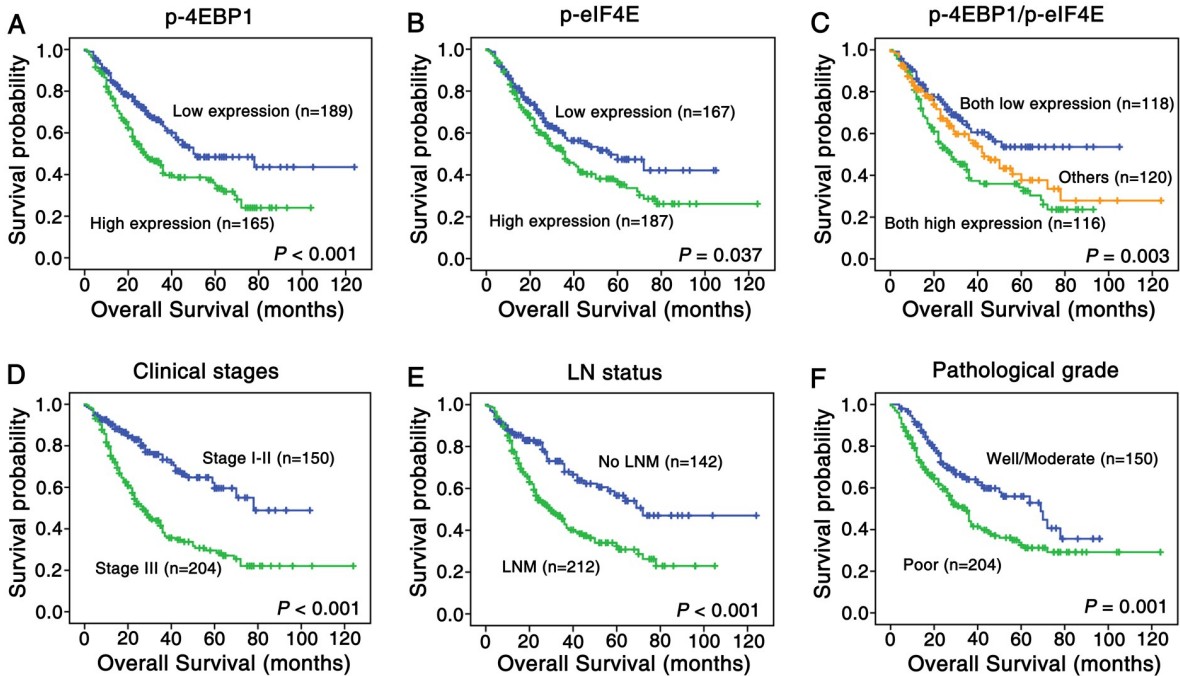

**Fig 3. Kaplan-Meier cures for overall survival of lung SCC and ADC patients with expression of p-4EBP1, p-eIF4E.** (A) NSCLC patients with high expression of p-4EBP1 showed worse overall survival rates compared to patients with low p-4EBP1 expression ($P = 0.001$, two sided). (B) NSCLC patients with high p-eIF4E expression showed worse overall survival rates compared to patients with low p-eIF4E expression ($P = 0.037$, two sided). (C) NSCLC patients with combined high expression of p-4EBP1 and p-eIF4E had worse overall survival rates than these with others ($P = 0.003$, two sided). (D) NSCLC patients with stage III owned poorer prognosis compared with that with stage I-II($P < 0.001$, two sided). (E) NSCLC patients with LNM had lower overall survival rate than those without LNM ($P < 0.001$, two sided). (F) NSCLC patients with well and moderate differentiation had higher overall survival rate than those with poor differentiation ($P = 0.001$, two sided).

was, the lower the survival rate ($P = 0.001$, Fig 3F & Table 4). In addition, there was no statistical significance between prognosis and age, gender, and histological type (Table 4).

We further investigated whether the overexpression of p-4EBP1 and/or p-eIF4E could be independent prognosis factors for NSCLC patients via multivariate analysis using cox regression method. Table 4 showed that expression of p-4EBP1 ($P = 0.013$), as well as clinical stage ($P < 0.001$), LNM ($P = 0.031$) and pathological grade ($P = 0.022$) were independent prognostic factors for NSCLC patients.

## Discussion

In this study, we found that p-4EBP1 and p-eIF4E expression in lung SCC and ADC patients were increased significantly compared to Non-CLT. Whether in lung SCC or ADC, p-4EBP1 had a positive relationship with p-eIF4E, and higher p-4EBP1 expression appeared in advanced clinical stage. Our analysis showed that NSCLC patients with high expressed p-4EBP1 and p-eIF4E alone or in combination had a lower survival rate than other expression patterns. Taking together, these data suggest high expressed p-eIF4E and p-4EBP1 could be new prognostic marker for NSCLC. Several research pointed out that overexpression of p-4EBP1 has been found in a range of other tumors, including ovarian cancer, hepatocellular carcinoma, and breast cancer, which confirmed their important role in various cancers [14,22,23]. In addition, 4EBP1 is associated with prognosis of patients in several kinds of cancers, such as renal cell cancer and small cell lung cancer [15,16]. Our data showed that the

**Table 4. Summary of univariate/multivariate analysis for overall survival in patients with NSCLC (n = 354).**

| Variables | Univariate analysis | | | Multivariate analysis | | |
|---|---|---|---|---|---|---|
| | Average survival time (SE) | 95%CI | P-value | Exp (B) | 95.0%CI | P-value |
| **p-4EBP1** | | | | | | |
| High expression | 45.631(3.395) | 38.977–52.285 | 0.000* | 1.524 | 1.092–2.126 | 0.013* |
| Low expression | 70.962(4.871) | 61.416–80.509 | | | | |
| **p-eIF4E** | | | | | | |
| High expression | 53.896(4.125) | 45.811–61.982 | 0.037* | 1.129 | 0.806–1.582 | 0.481 |
| Low expression | 60.673(4.137) | 52.564–68.783 | | | | |
| **Clinical stages** | | | | | | |
| Stage $_{I-II}$ | 70.370(4.204) | 62.131–78.609 | 0.000* | 2.016 | 1.369–2.968 | 0.000* |
| Stage $_{III}$ | 47.128(3.755) | 39.768–54.488 | | | | |
| **LN status** | | | | | | |
| LNM | 45.590(3.174) | 39.369–51.811 | 0.000* | 1.515 | 1.039–2.208 | 0.031* |
| No LNM | 76.287(5.436) | 65.632–86.943 | | | | |
| **Histological type** | | | | | | |
| SCC | 67.190(5.082) | 57.231–77.150 | 0.181 | 1.301 | 0.933–1.814 | 0.121 |
| ADC | 49.984(3.347) | 43.423–56.544 | | | | |
| **Pathological grade** | | | | | | |
| Well/moderated | 59.029(3.608) | 51.956–66.102 | 0.001* | 1.466 | 1.056–2.035 | 0.022* |
| Poor | 53.383(4.033) | 45.478–61.287 | | | | |
| **Age** | | | | | | |
| ≤50 | 45.334(3.648) | 39.182–53.485 | 0.632 | 1.099 | 0.781–1.546 | 0.598 |
| >50 | 61.368(3.947) | 53.631–69.105 | | | | |
| **Gender** | | | | | | |
| Female | 53.027(4.372) | 44.457–61.597 | 0.342 | 0.848 | 0.578–1.244 | 0.399 |
| Male | 58.892(3.783) | 51.478–66.306 | | | | |

Abbreviations: CI, confidence interval; Exp(B), odds ratio; SE, standard error; LNM, lymph node metastasis.

* $P < 0.05$.

survival time of patients with high p-4EBP1 is shorter than patients with low expressed p-4EBP1, and p-4EBP1 could be used as an independent prognostic indicator of NSCLC, regardless of clinical stage, LNM, and pathological grade. These results suggest that p-4EBP1 might be a potential prognostic indicator.

Eukaryotic initiation factors (eIFs) consisting of eIF4E, eIF2, eIF6 and so on are the main regulators in the initial stage of mRNA translation, whose abnormal expression has been confirmed to be involved in tumor progression, including proliferation, anti-apoptosis, and metastasis [24]. Our results showed that the phosphorylation level of eIF4E was significantly increased in lung SCC and ADC, especially ADC, and high expression of p-eIF4E was associated with poor prognosis in NSCLC patients, which is consistent with other researches founded in astrocytoma and nasopharyngeal carcinoma [25,26]. Besides eIF4E, other eIFs such as eIF2β [27], eIF3b [28], eIF3m [29], eIF4G1 [30] and eIF6 [31] are also validated abnormal activity in NSCLC. For instance, eIF2β is significantly up-regulated in NSCLC tissue and has a poor prognosis in patients with lung ADC. Knockdown of eIF2β inhibits the growth of lung ADC cells by inducing G1 cell cycle arrest [27]. EIF6 blocks the binding of large and small ribosomal subunits and prepares for ribosome binding to mRNA [24]. Gantenbein et al found that the expression of eIF6 protein in NSCLC was higher than that in healthy lung tissue, which was positively correlated with poor prognosis. The deletion of eIF6 can lead to cell

proliferation inhibition, caspase 3-mediated apoptosis, and ribosomal 60s maturation defects [31]. These confirm the importance of eIFs in tumorigenesis and development of NSCLC, and prove the value and targeted therapy strategies of p-eIF4E in NSCLC management.

Our data also showed a positively association between p-4EBP1 and p-eIF4E in lung SCC, as well as ADC. Moreover, patients with high p-4EBP1 and p-eIF4E expression had a lower OS rate compared to other expression patterns of the two proteins. The results suggest that there might be a crosslink between mTORC1 and MNKs pathways. Since eIF4G bring MNKs and eIF4E together to enhance MNK-mediated phosphorylation of eIF4E, eIF4E phosphorylation can be promoted by the phosphorylation of 4EBP1 which prevent eIF4E from binding with eIF4G [32,33]. Indeed, extracellular-regulated kinase (Erk) is a common upstream regulator for both MNKs and mTORC1 pathways [34]. Besides activating MNKs to enhance eIF4E phosphorylation, Erk also mediate the activation of mTORC1 pathway through phosphorylation of tuberous sclerosis complex 2 (TSC2) on multiple sites to promote phosphorylation of 4EBP1 [34]. Erk is known to be involved in development of numerous cancers including NSCLC [35–37]. In addition, there are several clues suggesting MNKs might cooperate with mTORC1 pathway. One study has shown that inhibition of MNKs by CGP57380 can enhance the inhibitory effect of rapalog on 4EBP1 phosphorylation [38]. The elucidation of additional pathway(s) linking mTORC1 and MNKs pathways could provide a new idea for the targeted treatment of NSCLC.

Cancer requires elevated protein synthesis to meet the increased metabolic demand. P-4EBP1 and p-eIF4E are downstream regulators of mTORC1 pathway and MNKs pathway of which the convergence node is eIF4E [11,17]. The research on mice with loss of eIF4E gene function shows that 50% reduction of eIF4E is compatible with normal development of the body and has a significant inhibitory effect on KRas driven lung cancer initiation [10]. In addition, mice lacking MNKs are reported to be viable and p-eIF4E is not essential for growth and development [39,40]. These findings suggest that inhibiting production of p-eIF4E and p-4EBP1 is a safe option for cancer treatment. Up to now, there are many mTOR inhibitors and MNKs inhibitors applied in cancer therapy, but the anticancer efficacy is not very ideal [41,42]. Although several studies have shown that mTOR inhibitors can suppress the production of p-4EBP1, it enhances eIF4E phosphorylation and leads to drug resistance. Moreover, combination of MNKs inhibitors can significantly reduce eIF4E phosphorylation induced by mTOR inhibitors [43]. Therefore, we infer that the interplay between mTORC1 pathway and MNKs pathway maybe a possible cause [44,45], which suggests that combined inhibition of p-4EBP1 and p-eIF4E could be a better regimen than single inhibition. Of course, combination therapy is still a growing field and its application and internal mechanism in NSCLC still need further study.

## Conclusion

In summary, p-4EBP1 and p-eIF4E might be novel prognostic markers for NSCLC, who possess potential application value for NSCLC targeted therapy.

## Author Contributions

**Data curation:** Sile Liu.

**Funding acquisition:** Songqing Fan, Qiuyuan Wen.

**Investigation:** Yaoxiang Tang, Jiadi Luo.

**Software:** Sile Liu, Hongmei Zheng.

**Writing – original draft:** Yaoxiang Tang, Yang Yang.

**Writing – review & editing:** Yuting Zhan, Qiuyuan Wen.

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
