## [Decision Letter · Decision Letter 0]

1 Dec 2021

PONE-D-21-09571Overexpression of p-4EBP1 associates with p-eIF4E and predicts poor prognosis for non-small cell lung cancer patients with resectionPLOS ONE

Dear Dr. Qiuyuan Wen,

Thank you for submitting your manuscript to PLOS ONE. After careful consideration, we feel that it has merit but does not fully meet PLOS ONE’s publication criteria as it currently stands. Therefore, we invite you to submit a revised version of the manuscript that addresses the points raised during the review process.

We look forward to receiving your revised manuscript.

Kind regards,

Johannes Haybaeck

Academic Editor

PLOS ONE

Journal Requirements:

The work was supported by grants of National Natural Science Foundation of China (grant No. 81703009, 81773218, and 81972838) and The Natural Science Foundation of Hunan Province (grant No. 2017JJ3457).

7. Your ethics statement should only appear in the Methods section of your manuscript. If your ethics statement is written in any section besides the Methods, please delete it from any other section. 

8. We note you have included a table to which you do not refer in the text of your manuscript. Please ensure that you refer to Table 4 in your text; if accepted, production will need this reference to link the reader to the Table.

Additional Editor Comments:

Based on the reviewer´s comments and based on the editor´s suggestions a major revision is required. In addition to the comments from referee one also the work by Gantenbein Nadine et al on eIF6 in lung cancer should be discussed.

Reviewers' comments:

Reviewer's Responses to Questions

**Comments to the Author**

1. Is the manuscript technically sound, and do the data support the conclusions?

Reviewer #1: Partly

2. Has the statistical analysis been performed appropriately and rigorously? 

Reviewer #1: Yes

3. Have the authors made all data underlying the findings in their manuscript fully available?

Reviewer #1: Yes

4. Is the manuscript presented in an intelligible fashion and written in standard English?

Reviewer #1: Yes

5. Review Comments to the Author

Reviewer #1: This study investigated that the potential prognostic significance of p-4EBP1 and phospho-eIF4E in total 354 NSCLC patients by immunohistochemistry. The expression of p-4EBP1 was associated with poor prognosis and was an independent poor prognostic factor. The results were basically well written; however, the reviewer has several comments.

1. Smoking status, PD-L1 expression level, and the information of oncogenic driver alterations, such as EGFR mutation and ALK fusion, would be useful to show the importance of p-4EBP1 and phospho-eIF4E expression levels in clinical field.

2. A variety of previous studies already showed the association of several eukaryotic initiation factors (eIF) with tumor progression and chemo-resistance. Therefore, to emphasize the clinical importance of p-4EBP1 and phospho-eIF4E expression in NSCLC, analyzing the disease-free survival would be informative because most of cases could have operability.

3. In table2, sample number of pathological grades ‘Well’ is too small to use Chi-square test. The reviewer recommends to analyze them by Fisher-exact test.

4. The abnormal activity of eIF complexes triggered by upstream signaling pathways is detected in many tumors, and eIFs can be a promising therapeutic target for various types of cancers. In NSCLC, eIF2β and eIF6 were shown as prognosis indicators and promising therapeutic targets (Cancer Sci. 2018 Jun;109(6):1843-1852. Eur J Cancer. 2018 Sep; 101:165-180.). The reviewer highly recommends to refer these papers and discuss the importance of the abnormal activity of eIF complexes in NSCLC.

6. PLOS authors have the option to publish the peer review history of their article (what does this mean?). If published, this will include your full peer review and any attached files.

Reviewer #1: No

---

## [Author Response · Author response to Decision Letter 0]

28 Dec 2021

Dear Editors and reviewer,

Thank you very much for the timely review on our manuscript entitled “Overexpression of p-4EBP1 associates with p-eIF4E and predicts poor prognosis for non-small cell lung cancer patients with resection” [PONE-D-21-09571], we believe these comments would be of great help to improve the manuscript. We have carefully revised the manuscript according to these comments, our point-by-point response to these suggestions are as following:

Journal Requirements:

Response: We have ensured that our manuscript meets PLOS ONE's style requirements, including those for file naming.

Response: We have provided additional details regarding participant consent both in Chinese and English versions.

Response: We have completed the modification in the "Funding Information" section and have confirmed that the grant numbers are correct. The "Funding Information" section is described below:

The work was supported by National Natural Science Foundation of China (grant No. 81703009, 81773218, and 81972838), the Natural Science Foundation of Hunan Province (grant No. 2017JJ3457), Hunan Provincial Innovation Foundation for Postgraduate (grant No. CX20210373), and the Fundamental Research Funds for the Central Universities of Central South University (grant No. 2021zzts1045).

The work was supported by grants of National Natural Science Foundation of China (grant No. 81703009, 81773218, and 81972838) and The Natural Science Foundation of Hunan Province (grant No. 2017JJ3457).

Response: Thank you very much for your suggestion. We were so sorry to confuse the editor to make Funding Statement “The author(s) received no specific funding for this work” by mistake in the first submission. We have modified the "Funding Information" in the revised manuscript and made a detailed and correct description in Funding Statement in our cover letter. The Funding Statement is described below:

The work was supported by National Natural Science Foundation of China (grant No. 81703009, 81773218, and 81972838), the Natural Science Foundation of Hunan Province (grant No. 2017JJ3457), Hunan Provincial Innovation Foundation for Postgraduate (grant No. CX20210373), and the Fundamental Research Funds for the Central Universities of Central South University (grant No. 2021zzts1045).

Response: Thanks for your suggestion. We have described our changes to our Data Availability statement in our cover letter. The Data Availability statement is described below:

All data relevant to the study are included in the article.

Response: Thanks a lot for your help. We have updated our information and authenticated the pre-existing iD in Editorial Manager following your guides.

7. Your ethics statement should only appear in the Methods section of your manuscript. If your ethics statement is written in any section besides the Methods, please delete it from any other section. 

Response: We have completed the modification as required in the revised manuscript.

8. We note you have included a table to which you do not refer in the text of your manuscript. Please ensure that you refer to Table 4 in your text; if accepted, production will need this reference to link the reader to the Table.

Response: We have completed the modification as required in the revised manuscript.

Additional Editor Comments:

Based on the reviewer´s comments and based on the editor´s suggestions a major revision is required. In addition to the comments from referee one also the work by Gantenbein Nadine et al on eIF6 in lung cancer should be discussed.

Response: Thank you for your advice. We have revised the manuscript according to the editor´s suggestions and discussed the work by Gantenbein Nadine et al on eIF6 in lung cancer in detail in the revised manuscript.

Reviewer :

1. Smoking status, PD-L1 expression level, and the information of oncogenic driver alterations, such as EGFR mutation and ALK fusion, would be useful to show the importance of p-4EBP1 and phospho-eIF4E expression levels in clinical field.

Response: We thank the reviewer for the comments. The reviewer mentioned three modules here: smoking status, PD-L1 expression level and the information of oncogenic driver alterations, such as EGFR mutation and ALK fusion, which are useful to emphasis the importance of p-4EBP1 and phospho-eIF4E expression levels in clinical field.

Smoking status Just as the reviewer highlighted the importance of relationship between smoking and the expression of p-4EBP1 and p-eIF4E, accumulating evidence has indicated nicotine is associated with mTOR pathway activation. For example, nicotine can promote the proliferation of human papillomavirus (HPV)-immortalized cervical epithelial cells H8 cells by activating Akt/mTOR pathway and inducing 4EBP1 phosphorylation [1]. Furthermore, nicotine and tobacco carcinogens can quickly activate Akt and mediate carcinogenesis [2]. In the present study, we would have analyzed the relationship between smoking status and the expression of p-4EBP1 and p-eIF4E, unfortunately, the patient's smoking status was not included in the original follow-up data. We have been aware of this problem and following up one by one to supplement the relevant information, but it is inevitable deficient and has not been finished yet since our samples were collected pretty much long time ago. We’ll keep on going to complete the data and take smoking status into account when we establish new clinicopathological data in the future.

 PD-L1 expression level Our previous study has detected the expression level of PD-L1 in non-small cell lung cancer (NSCLC) and found that PD-L1 was highly expressed in NSCLC tissues, which is related to tumor lymph node metastasis and poor prognosis of patients [3]. At present, there is no research report on the association between the expression of PD-L1, p-4EBP1 and p-eIF4E. Thanks for the suggestion of the reviewer, we will focus on this new field and further explore the internal relationship between PD-L1, p-4EBP1 and p-eIF4E in the follow-up work.

The information about changes in carcinogenic drivers, such as EGFR mutation and ALK fusion In our current study, none of the patients had EGFR mutation and ALK fusion detected and targeted therapy. Firstly, EGFR gene mutation and ALK fusion mutation strategy has not been received great importance and popularity many years ago. Secondly, the majority patients from remote mountainous areas could not afford the expensive costs of inspection and targeted treatment. Nowadays, not only EGFR mutation and ALK fusion detection has become a part of routine testing, but also other relevant carcinogenic drivers such as ROS1, MET, BRAF, TP53 and so on have been suggested to be detected in our department. Above all, thanks a lot for the reviewers’ suggestion, which is very important and helpful for our follow-up research.

[1] Chen L et al. eIF4E is a critical regulator of human papillomavirus (HPV)-immortalized cervical epithelial (H8) cell growth induced by nicotine. Toxicology. 2019; 419:1-10.

[2] West KA et al. Rapid Akt activation by nicotine and a tobacco carcinogen modulates the phenotype of normal human airway epithelial cells. J Clin Invest. 2003;111(1):81-90.

[3] Zheng H et al. Co-expression of PD-L1 and HIF-1α predicts poor prognosis in Patients with Non-small Cell Lung Cancer after surgery. J Cancer. 2021;12(7):2065-2072.

2. A variety of previous studies already showed the association of several eukaryotic initiation factors (eIF) with tumor progression and chemo-resistance. Therefore, to emphasize the clinical importance of p-4EBP1 and phospho-eIF4E expression in NSCLC, analyzing the disease-free survival would be informative because most of cases could have operability.

Response: We thank the reviewer for the comments. Based on a meta-analysis, we know that cancer patients with lower p-4EBP1 expression had better 3-year and 5-year disease-free survival [1]. Clear cell renal cell carcinoma patients whose tumors stained positive for p-4EBP1 had a higher disease-free survival (DFS) compared to patients whose tumors were negative [2]. However, there are few reports on the relationship between p-eIF4E and disease-free survival. In our present study, considering that the endpoint of disease-free survival is difficult to record, and that patients with cancers often have complications, which will interfere with the judgment of disease-free survival rate, we did not analyze the relationship between the p-4EBP1, p-eIF4E and disease-free survival, but tended to analyze whether the two proteins related to the overall survival in NSCLC. Of course, We will further improve the follow-up data and analyze the association between the expression of these two proteins and disease-free survival rate in the future.

[1] Zhang T et al. Meta-analysis of the prognostic value of p-4EBP1 in human malignancies. Oncotarget. 2017;9(2):2761-2769.

[2] Campbell L et al. Phospho-4e-BP1 and eIF4E overexpression synergistically drives disease progression in clinically confined clear cell renal cell carcinoma. Am J Cancer Res. 2015;5(9):2838-48.

3. In table2, sample number of pathological grades ‘Well’ is too small to use Chi-square test. The reviewer recommends to analyze them by Fisher-exact test.

Response: We thank the reviewer for the suggestion. We have re-analyzed the relationship between p-4EBP1 and p-eIF4E protein expression and pathological grade by Fisher-exact test in the revised manuscript.

4. The abnormal activity of eIF complexes triggered by upstream signaling pathways is detected in many tumors, and eIFs can be a promising therapeutic target for various types of cancers. In NSCLC, eIF2β and eIF6 were shown as prognosis indicators and promising therapeutic targets (Cancer Sci. 2018 Jun;109(6):1843-1852. Eur J Cancer. 2018 Sep; 101:165-180.). The reviewer highly recommends to refer these papers and discuss the importance of the abnormal activity of eIF complexes in NSCLC.

Response: We thank the reviewer for the suggestion. We have studied those two papers and discussed the importance of the abnormal activity of eIF complexes in NSCLC in the revised manuscript. 

Sincerely yours,

Qiuyuan Wen, MD/Ph. D

Department of Pathology of the Second Xiangya Hospital, Central South University

139 Ren Min Road, Changsha, Hunan 410011, China

---

## [Decision Letter · Decision Letter 1]

3 Mar 2022

Overexpression of p-4EBP1 associates with p-eIF4E and predicts poor prognosis for non-small cell lung cancer patients with resection

PONE-D-21-09571R1

Dear Dr. Qiuyuan Wen,

We’re pleased to inform you that your manuscript has been judged scientifically suitable for publication and will be formally accepted for publication once it meets all outstanding technical requirements.

Kind regards,

Johannes Haybaeck

Academic Editor

PLOS ONE

---

## [Editor Report · Acceptance letter]

14 Jun 2022

PONE-D-21-09571R1 

Overexpression of p-4EBP1 associates with p-eIF4E and predicts poor prognosis for non-small cell lung cancer patients with resection 

Dear Dr. Wen:

I'm pleased to inform you that your manuscript has been deemed suitable for publication in PLOS ONE. Congratulations! Your manuscript is now with our production department. 

Kind regards, 

on behalf of

Dr. Johannes Haybaeck 

Academic Editor

PLOS ONE